# Perspective: Potential Impact and Therapeutic Implications of Oncogenic PI3K Activation on Chromosomal Instability

**DOI:** 10.3390/biom9080331

**Published:** 2019-08-01

**Authors:** Bart Vanhaesebroeck, Benoit Bilanges, Ralitsa R. Madsen, Katie L. Dale, Evelyn Lau, Elina Vladimirou

**Affiliations:** 1UCL Cancer Institute, University College London, 72 Huntley Street, London WC1E 6BT, UK; 2Centre for Cardiovascular Sciences, Queens Medical Research Institute, University of Edinburgh, 47 Little France Crescent, Edinburgh EH16 4TJ, UK

**Keywords:** PI 3-kinase, chromosomal instability, PI3K inhibitor, cancer, tumour evolution, centrosome

## Abstract

Genetic activation of the class I PI3K pathway is very common in cancer. This mostly results from oncogenic mutations in *PIK3CA*, the gene encoding the ubiquitously expressed PI3Kα catalytic subunit, or from inactivation of the PTEN tumour suppressor, a lipid phosphatase that opposes class I PI3K signalling. The clinical impact of PI3K inhibitors in solid tumours, aimed at dampening cancer-cell-intrinsic PI3K activity, has thus far been limited. Challenges include poor drug tolerance, incomplete pathway inhibition and pre-existing or inhibitor-induced resistance. The principle of pharmacologically targeting cancer-cell-intrinsic PI3K activity also assumes that all cancer-promoting effects of PI3K activation are reversible, which might not be the case. Emerging evidence suggests that genetic PI3K pathway activation can induce and/or allow cells to tolerate chromosomal instability, which—even if occurring in a low fraction of the cell population—might help to facilitate and/or drive tumour evolution. While it is clear that such genomic events cannot be reverted pharmacologically, a role for PI3K in the regulation of chromosomal instability could be exploited by using PI3K pathway inhibitors to prevent those genomic events from happening and/or reduce the pace at which they are occurring, thereby dampening cancer development or progression. Such an impact might be most effective in tumours with clonal PI3K activation and achievable at lower drug doses than the maximum-tolerated doses of PI3K inhibitors currently used in the clinic.

## 1. Introduction

In this section, we provide a general introduction to PI3K and chromosomal instability, and describe in Section 2 how deregulated PI3K activity can affect chromosomal instability.

### 1.1. Class I PI3Ks–PIK3CA Mutation and Amplification

Class I PI3Ks are lipid kinases that signal downstream of tyrosine kinases, G protein-coupled receptors and small GTPases such as Ras, cdc42 and Rac (Figure 1A) and convert the membrane-bound lipid phosphatidylinositol(4,5)bisphosphate (PI(4,5)P_2_) to phosphatidylinositol(3,4,5)trisphosphate (PI(3,4,5)P_3_; also known as PIP_3_). This lipid, together with its degradation product PI(3,4)P_2_, regulates downstream signalling cascades involving Akt/PKB, mTORC1/2 and other proteins, ultimately inducing anabolic metabolism, cell-cycle progression, migration and pro-survival functions [1,2]. PIP_3_ and PI(3,4)P_2_ are both substrates for dephosphorylation at the 3′ position by the PTEN lipid phosphatase, effectively antagonising PI3K function. PTEN is a tumour suppressor whose frequent inactivation in cancer disrupts the normal dampening of class I PI3K signalling [3].

Mammals have four class I PI3K catalytic isoforms (p110α, β, γ and δ), encoded by four distinct genes, *PIK3CA*, *PIK3CB*, *PIK3CG* and *PIK3CD* (Figure 1A). The class IA subset of catalytic PI3Ks (p110α, β and δ) occur in a heterodimeric complex with a p85 regulatory subunit, of which there are three separate genes (*PIK3R1*, *PIK3R2* and *PIK3R3*). This heterodimerisation keeps the p85/p110 complex in an inactive state, which can be relieved by p85 binding to cell surface receptors.

This perspective focuses on *PIK3CA*, the class I PI3K that is most frequently mutationally-activated and/or gene-amplified in solid tumours (Figure 1B). In haematological malignancies, *PIK3CA* mutation or amplification is extremely rare (<0.1% in lymphoid and myeloid malignancies; cBioportal, accessed June 2019), with a slightly higher frequency of these events in *PIK3CD* (1% in lymphoid and myeloid malignancies; cBioportal, accessed June 2019), which is almost exclusively constituted by mutations in *PIK3CD* in diffuse large B-cell lymphoma. No consistent overexpression of class I PI3K subunits has been observed in haematological malignancies.

Common ‘oncogenic’ hotspots of *PIK3CA* mutations are found in the helical (E542K, E545K) and C-terminal domains (H1047R) of p110α [7]. Oncogenic mutations in *PIK3CA* lead to a constitutive de-inhibition of the p110α/p85 complex [8]. Such de-inhibition of the p85/p110 complex can also be achieved by mutations in the genes encoding the p85 subunits [9].

The non-mutated, wild-type *PIK3CA* gene is also frequently amplified in cancer, such as in lung squamous carcinoma where this occurs as part of amplification of the 3q genomic locus, the most common genomic aberration in this cancer [10]. It is currently unclear whether amplification of *PIK3CA* on its own leads to enhanced signalling. Whereas expression of oncogenically-mutated *PIK3CA* in cell-based studies leads to PI3K pathway activation (as assessed by phosphorylation of Akt/PKB), this is most often not the case upon overexpression of wild-type *PIK3CA* (see, for example, [11,12]). Overexpression of wild-type *PIK3CA* did not transform chicken embryo fibroblasts, in contrast to oncogenically mutated *PIK3CA* [13]. It is therefore possible that additional alterations in the pathway are required in order for *PIK3CA* amplification to result in PI3K pathway activation. In this context, recent data in squamous lung cancer show that *PIK3CA* amplification frequently co-occurs with amplification of Akt2 on chromosome 19 [14]. Evidence for a possible dosage dependency of genetic PI3K pathway activation in cancer, including for *PIK3CA* mutation [15,16], has recently re-emerged [4]. This is in line with other studies showing pervasive selection in cancer for oncogenic mutant allele imbalance [17,18].

In the text below, we will refer to *PIK3CA* mutation or amplification as ‘*PIK3CA* activation’.

### 1.2. Chromosomal Instability, Whole-Genome Doubling and Aneuploidy-A Cellular Stress

#### 1.2.1. Chromosomal Instability (CIN)

Chromosomal Instability (CIN) is the dynamic process of chromosomal alterations occurring at an elevated rate, in the form of structural or numerical changes. Numerical CIN has been attributed to mitotic defects such as erroneous kinetochore–microtubule attachments, compromised centromere geometry, supernumerary centrosomes, impaired spindle assembly checkpoint, and sister chromatid cohesion [19]. Structural CIN has been linked to pre-mitotic processes such as replication stress, failures in DNA repair and defective telomere maintenance. It is manifested by the presence of duplicated, deleted or rearranged chromosome fragments, acentric fragments (centromere absent), dicentric chromosomes (two centromeres) and chromatin bridges, all of which compromise genomic integrity. The investigation of the possible links between pre-mitotic and mitotic defects is an active area of research [20,21]. Importantly, oncogenic activation of several signalling pathways has been shown to contribute to the induction of CIN by compromising the above cellular processes [22]. CIN is thought to allow cancer cells to adapt to selective pressures during tumour evolution, metastasis and treatment [23,24], due to simultaneous dosage changes in a large number of genes.

#### 1.2.2. Whole Genome Doubling (WGD)

Whole Genome Doubling (WGD), the duplication of the entire chromosome complement, is prevalent in cancer and a significant proportion of human tumours (over 50% for breast and lung) undergo WGD during development [25,26]. Mechanisms implicated in WGD include cytokinesis failure at the end of mitosis, endoreduplication and cell fusion [27]. In most cases, tetraploid cells are genomically unstable and accumulate numerical and/or structural chromosomal abnormalities, which can contribute to tumour heterogeneity and evolution. A stochastic model, based on the potency and chromosomal distribution of oncogenes and tumour suppressor genes, suggested that WGD represents a path to a commonly observed near-triploid state [28]. Recent analyses have provided strong evidence that WGD provides a buffering effect in cancer cells against the effects of gene or chromosome losses or deleterious mutations [29].

#### 1.2.3. Aneuploidy, Cellular Stress and the CIN Paradox

Aneuploidy, referring to an abnormal number of chromosomes in a cell, imposes cellular stress due to imbalanced gene expression and increased burden on protein turnover machinery. This can lead to impaired proliferation [30,31,32,33,34,35] and can also induce cell death or senescence, often dependent on the level of CIN [36,37,38]. Studies in yeast suggest that perturbation of the stoichiometries of protein complexes, caused by the presence/absence of additional chromosomes, contributes to the so-called proteotoxic stress response [39,40]. Mammalian cells with an abnormal karyotype have been shown to activate the process of autophagy and lysosome-mediated protein degradation, possibly to cope with this stress [34,35]. Aneuploid yeast and human cancer cells also exhibit a sustained hypo-osmotic-like response, characterised by plasma membrane stress and impaired endocytosis, with an ensuing remodelling of cellular metabolism and a dependency on ubiquitin-mediated endocytic recycling of nutrient transporters [40].

It is not clear whether autophagy is a consequence or a cause of chromosomal aberrations. Autophagy has been implicated in preventing chromosomal instability and associated tumourigenesis, with impaired autophagy-promoting gene amplification, chromosome instability and aneuploidy [41]. A recent study suggested that autophagy is responsible for the cell death occurring at replicative crisis [42], underlining a tumour-suppressive role for autophagy at an early stage of immortalisation and malignant transformation.

It has now also become clear that the rate of chromosome segregation errors occurring in cancer cells must be within a sustainable range to avoid cell death associated with extreme karyotypic changes [31,32]. This helps to explain the so-called CIN/aneuploidy paradox [31], whereby extreme CIN is tumour-suppressive. Aneuploidy is thus not cancer-promoting per se, and low levels of CIN are now viewed as more biologically relevant for driving tumour-evolution and adaptation. Of note, deregulation of the tumour suppressor *TP53*, including through loss of *TP53* expression or gain-of-function mutations in *TP53*, is a well-known tolerance mechanism towards aneuploidy [43,44,45,46].

### 1.3. Microtubules and the Mitotic Spindle

Microtubules are protein structures composed of α/β-tubulin heterodimers in a head-to-tail arrangement. They nucleate from centrosomes (see Section 1.4 below) and are involved in many cellular processes such as mitosis, intracellular transport and cell motility [47,48,49]. Microtubules are polarised structures with a fast-growing, highly-dynamic plus-end (the main site of elongation) and a minus-end, which is slow growing, stable and often anchored at the centrosome. Microtubules are characterised by dynamic instability, undergoing cycles of growth and shrinkage, and are regulated by a broad range of stabilising and destabilising factors [47,48,49].

Chromosome segregation requires that chromosomes form attachments to the plus-ends of parallel microtubule bundles via kinetochores (Figure 2A). These are large multimeric protein complexes associated with the centromere of chromosomes that regulate the dynamics of the associated microtubules and power chromosome motion. The microtubule–kinetochore interface is highly dynamic, with microtubules growing and shrinking at the site of attachment. The correct attachment of sister kinetochores to microtubules emanating from opposite centrosomes is essential for the fidelity of chromosome segregation. Increased microtubule stability and defects in correction mechanisms of erroneous microtubule–kinetochore attachments lead to chromosome mis-segregation during cell division [23,50]. Altered microtubule assembly rates within mitotic spindles also impact CIN in cancer cells [51].

### 1.4. Centrosomes

In proliferating cells, two complex microtubule structures called centrioles, embedded within a matrix of proteins known as the pericentriolar material (PCM), form the centrosome, which functions as the main microtubule-organising centre (MTOC). In many non-proliferating cells, centrioles migrate to the cell surface to template the assembly of a cilium.

A cell normally has one centrosome, which is duplicated once per cell cycle. Aberrant cellular centrosome numbers are associated with aneuploidy, CIN and cancer [52,53,54]. While multipolar divisions and the resulting aneuploidy often lead to non-viable progeny [55,56,57,58], many cells with supernumerary centrosomes either cluster them into two spindle poles, or selectively inactivate extra centrosomes to enable bipolar divisions [59,60,61]. Centrosome clustering efficiency depends on the cell type. For example, epithelial cells have low clustering efficiency and do not tolerate extra centrosomes [62]. E-cadherin loss leads to increased cortical contractility, which is sufficient to promote centrosome clustering, facilitating the proliferation and survival of cancer cells with extra centrosomes [62]. Importantly, supernumerary centrosomes and multipolar spindle geometry increase the probability of creating merotelic attachments (a single kinetochore attached to two spindle poles), which is a major mechanism of aneuploidy and chromosomal instability in cancer cells [57,63,64].

Recent findings have reinforced the importance of centrosome amplification in cancer initiation [64,65,66,67]. It is important to mention, however, that there are several human conditions with rampant centrosome amplification, for example, due to mutations in centrosomal genes leading to microcephaly, dwarfism and ciliopathies, all of which are without a discernible increase in cancer (see, for example, [68]; reviewed in [69,70,71]).

In addition to their function as a MTOC, centrosomes have also been postulated to act as signalling hubs that integrate and coordinate a range of signalling pathways. This hypothesis is based on the observation that many signalling components, including kinases and phosphatases, have been associated with centrosomes and spindle poles [72,73,74] (see also Table 1; Table 2).

## 2. PI3K Pathway Activation and CIN/WGD

### 2.1. PIK3CA/Akt

Several oncogenic signalling pathways ranging from Ras and Raf [86,87,88,89,90,91,92] to Rb, Notch and many others have been shown to contribute to the induction of CIN (reviewed in [22]). We here propose a new potential CIN-inducing pathway, mediated by *PIK3CA*-*AKT*, which may also include downstream *MTOR* signalling.

We recently reported that expression of the *PIK3CA*^H1047R^ mutant can lead to centrosome amplification (in mouse embryonic fibroblasts (MEFs), the MCF10A immortalised breast cell line and in mouse tissues) and increased in vitro tolerance to WGD (in MEFs) [93], indicating that PI3K activation might be involved in CIN (Figure 2B). Signalling pathways involved in *PIK3CA*^H1047R^-driven centrosome amplification include AKT, ROCK and CDK2/Cyclin E-nucleophosmin [93]. Overexpression of Akt/PKB has been shown to induce supernumerary centrosomes/aneuploidy in cell lines [94,95]. There is also some preliminary evidence in HeLa cells suggesting that mTOR hyperactivation may induce polyploidy [96].

It is of interest to note that ~30% of the phosphopeptides differentially-phosphorylated in *PIK3CA*^H1047R^ MEFs compared to control cells belong to proteins associated with ‘cytoskeleton, cell-cycle and centrosome function’ [97]. In addition, expression of a gene set that controls mitosis (microtubule and mitotic spindle regulators) was found to be enriched in *PIK3CA^H1047R^* MCF10A breast cancer cells [98]. Interestingly, this gene set was not enriched in MCF10A cells that had lost PTEN [98]. A study mapping the dynamic protein–protein interaction network within the core insulin signalling pathway in *Drosophila* also found significant enrichment for proteins involved in ‘centrosome duplication’ following 10 and 30 min of insulin stimulation [99].

Indirect evidence for a role for PI3K in centrosome biology was previously found in human HeLa and HCT116 cell lines, in which stable transfection of an oncogenic c-Met tyrosine kinase receptor induced centrosome amplification in a PI3K-dependent manner, independent of the MAPK pathway [100]. At the time, it was unclear whether these observations were artefacts of cell-based overexpression studies. Our demonstration that physiological induction of *PIK3CA*^H1047R^ (i.e., in the heterozygous state and expressed from the endogenous promotor) can induce centrosome amplification and aneuploidy [93] more firmly establishes this biological role of mutant PI3K. In fact, PI3K also controls tetraploidisation under physiological conditions: in rodents, modulation of insulin concentration, and consequently Akt/PKB activity, orients hepatocytes into a specific cell-cycle program, leading to the generation of binucleated tetraploid cells [101].

*PIK3CA* mutation is very common in breast cancer, where it appears to be an early, clonal event [93]. Moreover, in genome-doubled breast cancers, the majority of *PIK3CA* mutations precede the genome duplication event, with *PIK3CA* mutations showing a tendency to be mutually exclusive with mutations in *TP53* [93], a known tolerance mechanism towards genome doubling [43,44,45]. These data indicate a potential role of *PIK3CA* mutation as a tolerance mechanism for genome doubling in breast cancer, independent of the p53 pathway. Additional studies are required to determine whether oncogenic *PIK3CA* activation might function as a CIN inducer, as tolerance mechanism to CIN induced by other stimuli, or both.

### 2.2. PTEN

PTEN loss has been extensively linked to CIN, a biological role that is largely dependent on the scaffolding rather than the phosphatase function of PTEN [102], and therefore mostly independent of PI3K/Akt activity. Given that these PTEN effects can therefore not be modulated by pharmacological intervention targeting PI3K, these will not be described here. For an excellent review on the functions of PTEN in the maintenance of genome integrity, the reader is referred to [102].

## 3. Potential Molecular Mechanisms Underlying *PIK3CA*-Related CIN

It is possible that genetic *PIK3CA*-activation, due to its constitutive signalling (as opposed to transient PI3K signalling upon growth factor stimulation), leads to a deregulation of cell biological processes in such a way that these become cancer-promoting.

### 3.1. Cell-Cycle Block by Constitutive PI3K (Over) Activation?

PI3K/Akt signalling regulates the activity and expression of several key proteins involved in cell-cycle progression, including cyclin D and the cyclin-dependent kinase inhibitor proteins p21 and p27 (reviewed in [103,104,105]). PI3K activity oscillates during the cell cycle [106,107,108,109] suggesting that pathway signalling dynamics may be important for phenotypic regulation. Consistent with this notion, forced expression of constitutive PI3Kα alleles blocks efficient cell-cycle progression [110,111], which can even lead to cell death under serum-free conditions [110]. A study using single-cell analysis provided further evidence that high PI3K activity cannot be sustained, with cells exhibiting high Akt activation in the cell population undergoing senescence [112]. Mechanisms implicated in a PI3K-driven cell-cycle block include a lack of downregulation of the cyclin E-Cdk2 complex, which normally disappears after entry into S phase of the cell cycle [110], and an inability to increase the activity of the transcription factor FOXO1 downstream of Akt, which is necessary for cell-cycle completion [111]. The complexity of PI3K signalling dynamics is, however, poorly captured by conventional studies of the pathway. This has spurred the development of computational models, with a recent Boolean model predicting the mechanistic basis for cellular PI3K pathway oscillations as well as their impact on cytokinesis and cell death [113].

In the studies mentioned above [110,111,112], the long-term fate of cells undergoing cell-cycle blockade has not been investigated. Should these cells manage to exit this cell-cycle block, this might involve cytokinesis defects and chromosomal segregation errors, amongst others. Even if this occurred at low frequency in a cell population, if it were to take place continuously, this process could give rise to cell clones with altered genetic constellations. These cells could then seed or maintain cancer development, akin to the ability of drug-tolerant persister cells—a small fraction of the bulk cancer cell population—to serve as a latent reservoir of cells for treatment-resistant tumour growth [114,115].

At present, it is not clear whether the observed defects in cell-cycle progression are an artefact of forced overexpression of supra-physiologically active PI3K constructs, or whether this would also occur upon endogenous oncogenic *PIK3CA* activation. In this regard, we did not find evidence for cell-cycle blockade in MEFs expressing an endogenous heterozygous *PIK3CA*^H1047R^ allele [93]. Oncogenic activation of PI3K signaling has also been shown to elicit senescence in some cellular contexts [112,116,117] but not in others [93,118]. Other than being context-dependent, evidence has been presented that *PIK3CA* activation is a weaker inducer of senescence than oncogenic Ras [119].

However, there is increasing evidence that genetic PI3K pathway activation in cancer might be a graded event, similar to observations of oncogenic K-Ras [18], with the presence of more than one mutant copy of *PIK3CA* and/or genetic activation of other PI3K pathway components seen in established tumours [4]. It is tempting to speculate that cell-cycle aberrations and possibly CIN might be induced beyond a certain threshold of sustained high PI3K pathway activation. In other words, in addition to disruption of the duration of PI3K pathway activation, the strength of the signal might also be of importance in the context of CIN.

### 3.2. Impact of PI3K on Microtubules and the Mitotic Spindle

Growth factor-stimulated PI3K activation [120] or oncogenic *PIK3CA* activation [121] have been shown to lead to microtubule stabilisation, a phenomenon that when sustained, can compromise genome integrity. Indeed, stabilisation of kinetochore–microtubule attachments at prometaphase or metaphase, by depletion of the kinesin-13 microtubule depolymerases Kif2b or MCAK, prevents the release of misoriented microtubule attachments, leading to merotelic attachments and other segregation effects [50].

Molecularly, the CLASP microtubule-associated proteins have been shown to colocalise at microtubule plus-ends in a PI3K/GSK3β-dependent manner, resulting in microtubule stabilisation at the leading edge of motile fibroblasts [122]. Stathmin, a microtubule-destabilising protein whose activity is known to be reduced by phosphorylation on Ser-38 [123], is also of potential relevance in this context. Indeed, high phospho-Ser38-Stathmin is associated with PI3K pathway activation in aggressive endometrial cancer [124], with PI3K pathway inhibitors dampening Ser38 phosphorylation [125].

GSK3β is a kinase of potential relevance to the regulation of microtubule stabilisation by PI3K. This Ser/Thr kinase is constitutively-active in cells, but becomes inactivated upon phosphorylation by Akt [126]. In addition to its role in metabolic/insulin signalling, GSK3β is involved in microtubule regulation, such as through the phosphorylation of MAPs (MT-associated proteins), APC (anaphase-promoting complex) and CLASPs and, when active, generally decreases microtubule stability. It is possible that sustained GSK3 inactivation, due to constitutive PI3K/Akt activity upon *PIK3CA* activation, contributes to microtubule stabilisation. Although early work linked GSK3β and microtubules upon transient PI3K/Akt stimulation by growth factors [77,127,128], GSK3β has since been largely ignored in the PI3K field.

PIP_3_ has been shown to be generated at the midcortex in metaphase cells and to be involved in spindle orientation [129,130,131]. The Carrera group further showed an involvement of p110α in these phenomena, with p110α becoming activated at mitosis entry and regulating early mitotic events, such as PIP_3_ generation, prometaphase progression and spindle orientation [131]. The latter could involve the regulation of cortical microtubule attachment complexes by p110α (Figure 2B). Indeed, in HeLa cells, inhibition of p110α dampens cortical clustering of LL5β, a microtubule-anchoring protein [132] that attaches EB1/CLASP-bound microtubule plus-ends to the cell cortex [133]. LL5β contains a PIP_3_-binding PH domain and is thus a putative downstream effector of p110α. Upon PI3K activation, LL5β in complex with CLASP relocalises to the plasma membrane and orientates stabilised microtubules towards the leading edge [133].

Of note, cytoplasmic PTEN has also been reported to be mainly associated with microtubules, which may be important in regulating its function [134].

It is worth pointing out the existence of conflicting data on the impact of PI3K-pathway mutation on cellular sensitivity to microtubule-deregulating anti-mitotic drugs (on their own or in combination with PI3K inhibitors) [11,120,135]. Many studies used early generation pan-PI3K/mTOR inhibitors and cancer cell lines with complex genetic backgrounds, precluding a clear interpretation of the observations made. In addition, several studies have used the PI3K inhibitor BKM120/Buparlisib, which directly binds to microtubules and interferes with microtubule polymerisation [136,137].

### 3.3. Impact of PI3K on Centrosomes (and Vice Versa)

Recent data have implicated PI3K activation in centrosome amplification, such as by *PIK3CA*^H1407R^ activation in MEFs [93]. Centrosome amplification induced by the oncogenic c-Met-tyrosine kinase receptor has been shown to occur via PI3K/Akt, independent of the MAPK pathway [100]. Moreover, EGF controls centrosome separation through Akt, independent of MEK, MAPK or mTORC1 pathways [138].

*PIK3CA* activation is known to promote a mesenchymal morphology in a range of epithelial cell lines [117,139]. Given that epithelial cells are innately inefficient at centrosome clustering [62], such *PIK3CA*-induced epithelial-to-mesenchymal transition and altered cortical tension may enable these cells to efficiently cluster supernumerary centrosomes [62] and divide in the wake of misaligned chromosomes, pre-disposing them to CIN. On the other hand, persistent proliferation of cells undergoing epithelial-to-mesenchymal transition has been shown to lead to genomic instability [140].

Interestingly, several PI3K pathway components (Table 1) and substrates for the PI3K-activated kinase Akt/PKB (Table 2) can be found in centrosomes. Members of the transforming acidic coiled-coil (TACC) family of proteins, implicated in centrosome regulation and cancer, are of particular interest in this context [141,142]. Indeed, TACC1, a centrosome-localised microtubule-binding protein, was recently identified as a top hit in a genome-wide shRNA screen for genes that convert PI3K inhibitor-induced cytostasis to cytotoxicity, selectively in *PIK3CA-*mutant cells [143]. This is in line with data indicating that TACC1 overexpression protects cells from cytostasis induced by PI3K inhibition [144]. This differential sensitivity may relate to a vulnerability generated by microtubule/centrosome changes in *PIK3CA*-mutant cells. Moreover, the TACC3 isoform interacts with TSC2 [145], a bona fide Akt substrate, suggesting a possible interplay between PI3K/Akt and TACC/centrosome pathways. FGFR-TACC fusion proteins, generated by gene translocations in cancer, have also been shown to display oncogenic activity, to induce chromosomal segregation defects and to trigger aneuploidy [146].

Several other studies support a reciprocal relationship between the PI3K pathway and centrosomal regulation. An RNAi screen in *Drosophila* thus identified proteins involved in the centrosome cycle as regulators of acute insulin-dependent AKT activation [99]. Microtubule acetylation has also recently been linked to sequestration and inhibition of AKT [147]. Further understanding of this intricate crosstalk is required to determine how it may influence or be influenced by oncogenic *PIK3CA*-induced centrosome amplification.

## 4. Potential Therapeutic Exploitation in Cancer Prevention and Dampening of Tumour Evolution

At present, there are no therapies aiming to interfere with tumour evolution [23]. As argued above, it is possible that the generation and/or tolerance of CIN is a cancer-promoting activity of oncogenic PI3K activation, resulting in irreversible genetic alterations and a greater ability to adapt to selective pressures, due to clonal diversification [23]. This potentially pre-disposes cells to aneuploidy, setting the scene for cooperation with other cancer-promoting genetic lesions and fostering cell-to-cell variation, the substrate for tumour evolution. This would potentially allow *PIK3CA* activation to act as a continuous driver/facilitator of genetic changes in cancer, but also offer a druggable opportunity to interfere with tumour evolution. An important question is which cancers would potentially benefit from interfering with PI3K activity to block CIN, and whether such an approach would require the maximum-tolerated doses of PI3K inhibitors that are currently used in the clinic.

With regards to the first question, it is of relevance to note that mutation of *PIK3CA* is an early clonal event, occurring before WGD, in both breast cancer [93] and in colon cancer [148]. This is also the case for *PIK3CA* amplification (as part of a 3q arm amplification) in lung squamous cell carcinoma [149]. In some cases of colon cancer, *PIK3CA* mutation was estimated to have occurred at least a decade before cancer diagnosis [148]. Especially upon early diagnosis of the presence of such genetic alterations, for example, using liquid biopsies that can detect *PIK3CA* mutation [150], it is tempting to speculate that administration of low doses of PI3K pathway inhibitors could be used as a strategy to prevent cancer development and/or progression [151].

In other cancer types where PI3K pathway activation is not clonal such as in renal cell carcinoma, evidence indicates that the PI3K–AKT–mTORC1 pathway often becomes recurrently activated at multiple branch points during tumour evolution [152]. This occurs in separate subclones within the same tumour [152] and is an example of so-called parallel tumour evolution that converges upon PI3K activation [153]. In these cancers, interference with PI3K signalling as part of a maintenance therapy might also dampen tumour evolution.

With regards to the question of drug dose, it would be challenging to administer high doses of PI3K inhibitors in a preventative or maintenance setting, due to toxic side effects and cost. Moreover, high doses of PI3K inhibitors aiming to wipe out all cellular PI3K activity often give rise to feedback PI3K re-/over-activation, which would clearly be undesirable in this context. Although this remains to be proven, it is possible that oncogenic *PIK3CA* leads to low-level PI3K signalling, compared to growth-factor-stimulated PI3K activation. This would mean that low doses of PI3K inhibitors might allow the specific dampening of oncogenic PI3K signalling while still allowing normal PI3K signalling, and thereby interfere with tumour evolution. The feasibility of such low-dose PI3K inhibitor treatment is supported by observations in patients with so-called *PIK3CA*-related overgrowth syndrome (PROS), caused by embryonic acquisition of mosaic activating *PIK3CA* mutations [154]. In this context, long-term administration of low doses of a PI3Kα-selective inhibitor have led to dramatic improvements in disease severity with negligible or no side effects [155], with some patients having been on a low dose PI3Kα inhibitor for up to 3.5 years (Guillaume Canaud, *personal communication*).

It is important to emphasise that administration of such a low dose of a PI3K inhibitor is not expected to induce anti-proliferation or cancer regression in advanced cancers, which would require standard anti-cancer therapies such as chemotherapy or surgery.

## 5. Concluding Remarks

In this perspective, we have summarised the evidence for a potential role of *PIK3CA* activation in the induction and/or tolerance of CIN. Such a role could be indirect; for example, by providing survival signals to cope with the cell stress imposed by CIN. A role in CIN could also derive from the sustained pathway stimulation by *PIK3CA* activation (which might disrupt cellular processes that normally require cyclical regulation such as cell-cycle progression) and/or the dose level of PI3K pathway activation.

A clear role of *PIK3CA* in CIN may not have been uncovered to date because PI3K research has mainly focused on the analysis of bulk cell populations, using either PI3K inhibitors, experimental conditions of acute signalling often by supra-physiological doses of growth factors, and established *PIK3CA* mutant cells that have adapted to constitutive PI3K activation. It is clear that single cell analyses will be critical to capture possible CIN and cell heterogeneity induced by *PIK3CA* activation, in addition to appropriate cell- and tissue-based models such as 2D/3D cultures [156] and inducible models of *PIK3CA* activation. Bioinformatic analysis of human tumours may also allow the correlation of the timing of *PIK3CA* activation with CIN in whole tissues. Together, such data could underpin the potential exploitation of this knowledge in cancer prevention, or dampening cancer progression using PI3K pathway inhibitors. Given that other growth-factor-related oncogenes such as *Ras* and *Raf* might also regulate CIN [86,87,88,89,90,91,92], one could even speculate that a cocktail of oncogene pathway inhibitors, given at low doses, might be an effective way to dampen cancer evolution (reviewed in [151]).

## Figures and Tables

**Figure 1 biomolecules-09-00331-f001:**
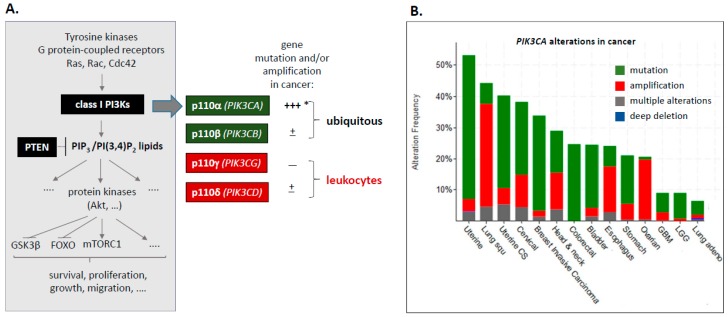
(**A**) Overview of class I PI3K signalling and genetic alterations in PI3K catalytic subunits in cancer. * Note that both wild-type and mutant *PIK3CA* alleles can be amplified in cancer [4]. A recent analysis showed that *PIK3CA*-mutant cancers frequently have more than one copy of the mutated *PIK3CA* gene and/or harbour a second *PIK3CA* variant [4]. (**B**) Frequency of *PIK3CA* alterations in a range of human cancers. Data are from cBioportal [5,6] (accessed June 2019).

**Figure 2 biomolecules-09-00331-f002:**
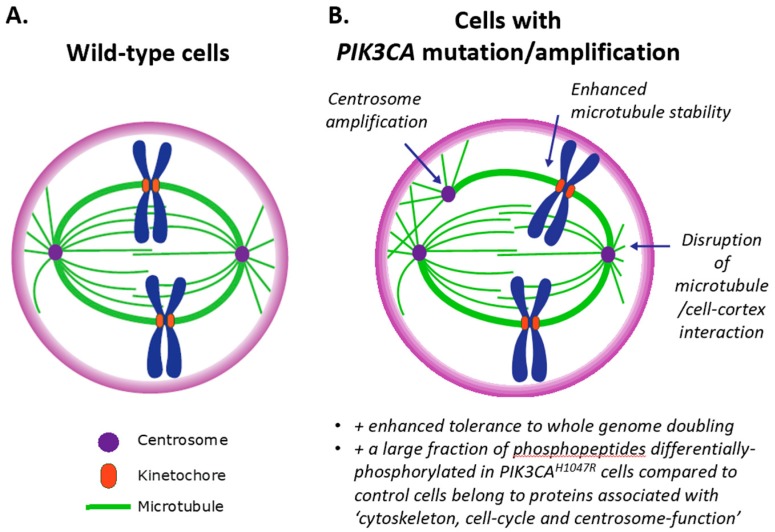
Schematic depicting the mitotic spindle in wild-type cells (**A**) and how this is deregulated in cells with PIK3CA mutation/amplification (**B**).

**Table 1 biomolecules-09-00331-t001:** PI3K pathway components found in centrosome/primary cilium.

PI3K Pathway Component	References	Additional Information
p85 regulatory subunit of PI3K	[75,76]	p85 can associate with the centrosome in an insulin-dependent manner
Akt/PKB	[77]	Akt/PKB is phosphorylated during mitosis and is present in the centrosome
	[78]	T308-phosphorylated Akt/PKB is present in basal body of primary cilia
	[79]	S473-phosphorylated Akt/PKB is present in basal body of primary cilia. Akt/PKB interacts with and phosphorylates the ciliary protein Inversin.
	[80]	Akt/PKB-inhibition prevents recruitment of PTEN to mitotic centrosomes
TSC1/TSC2 (TSC2 is an Akt/PKB substrate)	[81]	TSC1 is present in centrosome. Phosphorylated TSC1 and phosphorylated TSC2 co-immunoprecipitate with Plk1
GSK3β (Akt/PKB substrate)	[77]	Phospho-GSK-3 at the centrosomes upon entry into mitosis
PTEN	[82]	Phosphatase-independent (scaffold function) of PTEN is recruited to pre-mitotic centrosomes in a Plk1-dependent fashion

**Table 2 biomolecules-09-00331-t002:** Akt/PKB substrates found in centrosome/primary cilium.

Akt/PKB Substrate	References	Additional Information
TSC1/TSC2 (TSC2 is an Akt/PKB substrate)	[81]	TSC1 is present in centrosome. Phosphorylated TSC1 and phosphorylated TSC2 co-immunoprecipitate with Plk1
GSK3β	[77]	Phospho-GSK3 at the centrosomes upon entry into mitosis
Inversin	[79]	Akt/PKB interacts with and phosphorylates the ciliary protein Inversin—dimerisation. Co-localisation of Inversin and phosphorylated-Akt/PKB at the basal body is augmented by PDGF-AA.
TEIF	[83]	TIEF is a potential centrosome component
Girdin	[84,85]	Girdin may localise to centrosomes

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
