# Peer review of "Perspective: Potential Impact and Therapeutic Implications of Oncogenic PI3K Activation on Chromosomal Instability"

_biomolecules, 2019, doi:10.3390/biom9080331_

Round 1
Reviewer 1 Report
The review by Vanhaesebroeck et al. describes the association of the PI3K signaling in cancerogenesis. Authors describe the PI3K pathway, the activated PI3K signaling in cancer and its involvement in regulation / misregulation of genome stability.
The review is nicely written and covers important aspects including potential therapeutics. However, it is suggested to re-organize the review.
Authors describe under points 1.2. to 1.4. generally whole-genome instability, microtubule organization in mitosis, centrosome function without a context for the association with PI3K, which is important for the general reader.
Major points:
1. Please integrate PI3K activity into the points 1.2. to 1.4.
2. Authors describe very often an association. For some experiments there are however causative data available that go beyond just an association of observations. Some experiments reveal a mechanistic causative link between PI3K activity and a changed pathway, such as by knock-down experiments or usage of the activated PI3K mutant that more directly link the PI3K to observed changes in tumors.
Author Response
London, 27 July 2019
We thank the Referees for their positive feedback on our manuscript, and the high scores provided on the different aspects of the review. Please see below how we have addressed their comments.
We trust that our Review will now be acceptable for publication.
On behalf of the authors,
Bart Vanhaesebroeck
Referee 1
1. The current structure of our review is to first provide a general introduction in Section 1 on the concepts of PI3K and CIN, before bringing these two concepts together in Section 2, i.e. how PI3K affects CIN. Section 1 provides generic concepts and background for the non-specialist reader, whereas the specialist reader can focus on Section 2. Referee 1 has asked us to combine these sections. An earlier version of our manuscript had such an amalgamation, which readers found long-winded and ‘not to the point’. We therefore prefer to keep the current clear split-up of sections. However, in order to address this Referee comment, we have now clarified the structure of the review at the outset (see lines 32-33: ‘In this section, we provide a general introduction to PI3K and chromosomal instability, and describe in Section 2 how deregulated PI3K activity can affect chromosomal instability’).
2. We have to admit that we do not understand this comment: Section 2 covers the requested information, and summarizes the mechanistic data to support a link between PI3K modulation and CIN.
Reviewer 2 Report
Review Report
biomolecules-566724 review article
Perspective: Potential impact and therapeutic implications of oncogenic PI3K activation on
chromosomal instability; Vanhaesebroeck, Bilanges, Madsen, Dale, Lau and Vladimirou
Vanhaesebroeck et al. have composed a comprehensive review on PI3K activation and chromosomal instability (CIN) in solid tumors. They conclude that inhibition of PI3K and other CIN inducing oncogenes such as Ras and Raf might be an effective way to dampen cancer evolution by preventing CIN induction. The feasibility of a low dose PI3K inhibitor therapy is discussed.
The content of the article appears to be adequate for the description of the stated phenomena in both the quality and amount of data.
Specific comments
Lane 55: This perspective focuses on PI3KCA in solid tumors. The reader may be interested to learn about the situation of PI3KCA and CIN in hematological malignancies. PI3KCA is overexpressed in haematological malignancies. Hematological malignancies also harbor CIN. Is there any evidence on a link between PI3KCA and CIN in hematological malignancies? This should be mentioned in a sentence with relevant references, or discussed in a more elaborate chapter including relevant references, e.g. Piddock et al., Cancers 2017, 9, 29; doi:10.3390/cancers9040029; The Role of PI3K Isoforms in Regulating Bone Marrow Microenvironment Signaling Focusing on Acute Myeloid Leukemia and Multiple Myeloma. Herschbein and Liesveld, Blood Reviews 32 (2018); Dueling for dual inhibition: Means to enhance effectiveness of PI3K/Akt/mTOR inhibitors in AML.
If the authors do not wish to include information on hematological malignancies, they should clarify the title: Potential impact and therapeutic implications of oncogenic PI3K activation on chromosomal instability in solid tumors.
Lane 123: The authors should specify the mechanisms of TP53 inactivation to induce tolerance of aneuploidy. Is a loss of TP53 expression sufficient? Is TP53 mutation required?
Lane 176: The authors should mention the CIN inducing pathways including Rb, Wnt, TP53, Ras, Notch, TGFb, NFkB, Integrin, Hippo, and clarify that they propose an additional CIN inducing pathway: PI3KCA-AKT-mTor.
Author Response
London, 27 July 2019
We thank the Referees for their positive feedback on our manuscript, and the high scores provided on the different aspects of the review. Please see below how we have addressed their comments.
We trust that our Review will now be acceptable for publication.
On behalf of the authors,
Bart Vanhaesebroeck
Referee 2
1. Line 55 (line 58 in the revised MS): This Referee refers to the sentence: ‘This perspective focuses on PIK3CA, the class I PI3K that is most frequently mutationally-activated and/or gene-amplified in solid tumours (Figure 1B)’. It is correct that we should have elaborated on a possible role of class I PI3K deregulation in haematological malignancies as well. We have now addressed this in the revised text as follows (lines 59-64): ‘In haematological malignancies, PIK3CA mutation or amplification is extremely rare (< 0.1% in lymphoid and myeloid malignancies; cBioportal, accessed June 2019), with a slightly higher frequency of these events in PIK3CD (1% in lymphoid and myeloid malignancies; cBioportal, accessed June 2019), which is almost exclusively contributed by mutations in PIK3CD in Diffuse Large B-Cell Lymphoma. No consistent overexpression of class I PI3K subunits has been observed in haematological malignancies.’ This Referee also mentions that PIK3CA is overexpressed in haematological cancers. To the best of our knowledge, this is not the case. Given the uncertainty over PI3K alterations in haematological malignancies, we have decided not to alter the title to include reference to solid tumours.
2. Line 123 (lines 131-133 in the revised MS): ‘Of note, inactivation of the tumour suppressor TP53 is a well-known tolerance mechanism towards aneuploidy [41-43]’. The Referee asks if loss of TP53 expression is sufficient or whether TP53 mutation is required. This is a good point, and we have now included an additional reference (J Cell Biochem. 2012 Feb;113(2):433-9. doi: 10.1002/jcb.23400 - Links between mutant p53 and genomic instability) and updated this sentence as follows: (lines 131-133 in the revised manuscript): ‘Of note, deregulation of the tumour suppressor TP53, including through loss of TP53 expression or gain-of-function mutations in TP53, is a well-known tolerance mechanism towards aneuploidy [43-46].’
3. Line 176 (lines 186-187 in the revised manuscript): The Referee refers to the sentence ’Several oncogenic signalling pathways have been shown to contribute to the induction of CIN [20]’ and asks us ‘to mention the CIN inducing pathways including Rb, Wnt, TP53, Ras, Notch, TGFb, NF-kB, integrin, Hippo, and clarify that they propose an alternative CIN-inducing pathway: PIK3CA-AKT-mTOR’. These pathways (and many more) are all mentioned in Figure 1 in the cited Reference 20 (Orr, B. and D.A. Compton, A double-edged sword: how oncogenes and tumor suppressor genes can contribute to chromosomal instability. Front Oncol, 2013. 3: p. 164) but we agree that it would be good to refer to the multiple pathways (without naming them all as suggested) that contribute to the induction of CIN, and be more explicit on what our review aims to contribute. We have therefore revised the text as follows: ‘Several oncogenic signalling pathways ranging from Ras and Raf [86-92] to Rb, Notch and many others have been shown to contribute to the induction of CIN (reviewed in Ref.[22]). We here propose a possible additional CIN-inducing pathway, mediated by PIK3CA-AKT, which may also include downstream MTOR.’ This revised sentence now also includes reference to Refs [86-92] which were previously only cited at the end of the manuscript.